# Survival Is Related to Estrogen Signal Transduction Pathway Activity in Postmenopausal Women Diagnosed with High-Grade Serous Ovarian Carcinoma

**DOI:** 10.3390/cancers13205101

**Published:** 2021-10-12

**Authors:** Laura van Lieshout, Phyllis van der Ploeg, Yvonne Wesseling-Rozendaal, Anja van de Stolpe, Steven Bosch, Marjolein Lentjes-Beer, Meggy Ottenheijm, Annelen Meriaan, Caroline Vos, Joanne de Hullu, Leon Massuger, Ruud Bekkers, Jurgen Piek

**Affiliations:** 1Department of Obstetrics and Gynecology, Catharina Cancer Institute, Catharina Hospital, P.O. Box 1350, 5602 ZA Eindhoven, The Netherlands; phyllis.vd.ploeg@catharinaziekenhuis.nl (P.v.d.P.); meggyottenheijm@gmail.com (M.O.); annelenmeriaan1993@gmail.com (A.M.); ruud.bekkers@catharinaziekenhuis.nl (R.B.); 2Department of Obstetrics and Gynecology, Radboud Institute for Health Sciences, Radboud University Medical Center, P.O. Box 9101, 6500 HB Nijmegen, The Netherlands; Joanne.deHullu@radboudumc.nl; 3GROW School for Oncology and Developmental Biology, Maastricht University, P.O. Box 616, 6200 MD Maastricht, The Netherlands; 4Molecular Pathway Diagnostics, Philips, High Tech Campus 11, 5656 AE Eindhoven, The Netherlands; yvonne.wesseling-rozendaal@philips.com (Y.W.-R.); anja.van.de.stolpe@philips.com (A.v.d.S.); 5Laboratory for Pathology and Medical Microbiology (Stichting PAMM), P.O. Box 2, 5500 AA Veldhoven, The Netherlands; S.Bosch@pamm.nl; 6Laboratory for Pathology, Jeroen Bosch Hospital, P.O. Box 90153, 5200 ME ‘s-Hertogenbosch, The Netherlands; ma.lentjes@jbz.nl; 7Department of Obstetrics and Gynecology, Elisabeth-TweeSteden Hospital, P.O. Box 90151, 5000 LC Tilburg, The Netherlands; c.vos@etz.nl; 8Radboud Institute for Health Sciences, Radboud University Medical Center, P.O. Box 9101, 6500 HB Nijmegen, The Netherlands; l.massuger@kpnmail.nl

**Keywords:** estrogen signaling, ovarian cancer, signal transduction pathways

## Abstract

**Simple Summary:**

All cells have a complex internal network of ‘communication chains’ called signal transduction pathways (STPs). Through interaction of different proteins in STPs, they are partly responsible for the behavior of a cell. In our study, we investigated the activity of six STPs in 85 women with advanced stage high-grade serous ovarian cancer (HGSC). To investigate the relation between and differences in survival and STP activity, women with a short disease-free survival (below 12 months) and a long disease-free survival (above 24 months) were included. We found no differences in mean STP activity between short-term survivors (52 women) and long-term survivors (33 women). However, when we analyzed postmenopausal women, we found that both disease-free and overall survival were related to estrogen receptor (ER) pathway signaling. This indicates that a better survival outcome was related to a more active ER pathway in this subgroup.

**Abstract:**

High-grade serous ovarian carcinoma (HGSC), the most common subtype of ovarian cancer, has a high mortality rate. Although there are some factors associated with survival, such as stage of disease, there are remarkable differences in survival among women diagnosed with advanced stage disease. In this study, we investigate possible relations between survival and signal transduction pathway (STP) activity. We assessed the functional activity of the androgen receptor (AR), estrogen receptor (ER), phosphoinositide-3-kinase (PI3K), Hedgehog (HH), transforming growth factor beta (TGF-β) and canonical wingless-type MMTV integration site (Wnt) pathway in 85 primary tumor samples of patients with FIGO stage IIIC to IVB HGSC and disease-free survival (DFS) below 12 (*n* = 52) or over 24 months (*n* = 33). There were no significant differences in median pathway activity between patients with a short and long DFS. In univariate Cox proportional hazards analysis, ER pathway activity was related to a favorable DFS and overall survival (OS) in postmenopausal women (*p* = 0.033 and *p* = 0.041, respectively), but not in premenopausal women. We divided the postmenopausal group into subgroups based on ER pathway activity quartiles. Survival analysis revealed that postmenopausal women in the lowest ER quartile had a shorter DFS and OS (log-rank *p* = 0.006 and *p* < 0.001, respectively). Furthermore, we were able to form subgroups of patients based on an inverse relation between ER and PI3K pathway activity. In conclusion, in postmenopausal patients with advanced stage HGSC, a poorer survival outcome was associated with low functional ER pathway activity.

## 1. Introduction

Ovarian cancer is the most lethal gynecological malignancy and the fifth leading cause of cancer related death in women [1]. High-grade serous ovarian carcinoma (HGSC) is the most common histotype of ovarian cancer and is often detected at an advanced stage of disease (i.e., FIGO stage IIIC-IVB) [2,3]. At this point, the overall five-year survival rate is less than 40%, even after successful first-line treatment with debulking surgery and chemotherapy [2]. Despite the generally poor prognosis, the range in both disease-free (DFS) and overall survival (OS) in patients diagnosed with advanced stage HGSC is remarkable, with some patients remaining disease-free for over a decade [4].

Several clinicopathological factors associated with improved survival have been established like stage of disease, CA125 concentration at diagnosis and after treatment, the possibility of primary debulking surgery and residual disease after surgery. Furthermore, immune factors such as tumor infiltrating lymphocytes have been identified as prognostic factors [5,6]. However, even in patients with poor prognostic factors, long-term survival is not uncommon [7]. Furthermore, in long-term survivors with recurrent disease, both short and long DFS times are seen; some patients recur swiftly but respond well to therapy while others have a long DFS with a short survival time after recurrence [5]. Despite well-established prognostic factors, a proportion of tumors intrinsically behaves more or less aggressively. Unfortunately, the assessment of tumor specific characteristics associated with survival in ovarian carcinoma is complicated by a large heterogeneity in genomic mutations. For many cancer types, a single identifying trait is found in a substantial number of patients, such as HER2 amplification which is present in 20 to 25% of breast cancer patients [8,9]. However, research aimed at the analysis of gene expression profiles and patterns to reveal a relation to survival, could not confirm this for ovarian carcinoma [7]. In addition, a focus on specific gene alterations in the genotype of cancer cells disregards the functional phenotype of cancer cells, while it is becoming increasingly clear that the functional phenotype is influenced by other factors, such as the tumor microenvironment.

In 2010, Verhaegh et al. developed a technique to quantitatively measure functional signal transduction pathway (STP) activity, and therefore the functional phenotype, of cancer cells. With this assay, mRNA levels of target genes of major oncogenic STPs are used as input for knowledge-based Bayesian network models [10,11,12,13]. In previous studies in breast and colon cancer, the accuracy of this assay in determining aberrant STP activity has been validated. For breast and endometrial cancer, ER pathway activity was related to prognosis and in breast cancer the assay was superior to traditional immunohistochemical staining in the prediction of response to tamoxifen treatment [10,14,15].

In this exploratory study, we use these pathway assays in patients diagnosed with advanced stage HGSC who achieved complete remission after treatment with debulking surgery and (neo-)adjuvant chemotherapy. To investigate disease recurrence and survival in relation to STP activity, we compare STP activity in patients with a short and long DFS as well as define interrelations between different pathways with regard to menopausal status and survival. We aim to explain the differences in survival and ultimately provide new leads for accurate selection of patients for targeted therapies.

## 2. Materials and Methods

### 2.1. Study Design and Population

We retrospectively searched the Netherlands Cancer Registration (NCR) database for patients diagnosed with FIGO stage IIIC and IV HGSC between January 2000 and December 2016 in three Dutch hospitals (Catharina Hospital Eindhoven, Elisabeth-TweeSteden Hospital Tilburg and Radboud University Nijmegen Medical Center). Patients were eligible for inclusion if histology of the primary tumor was taken prior to start of chemotherapy and available for analysis and if treatment with primary or interval debulking surgery combined with (neo-)adjuvant chemotherapy with carboplatin and paclitaxel resulted in complete remission. Complete remission was chosen as inclusion criterium as we aimed to compare DFS, which requires patients to achieve complete remission first. Patients were excluded if they objected to further use of pathology samples or if they were diagnosed with any other malignancy, either prior to or following HGSC diagnosis, with the exception of basal cell carcinoma as cases where this affects life expectancy are exceedingly rare. Based on reported median DFS for advanced stage HGSC of 16 and 19 months, we decided to exclude patients with a DFS between 12 and 24 months from our analysis to form two clearly defined groups [16,17]. As such, we hypothesized to be able to clearly identify possible differences in STP activity between short-term and long-term disease free survivors. Patients with a DFS below 12 months were defined as the ‘short DFS group’ and patients with a DFS above 24 months as the ‘long DFS group’.

### 2.2. Data Collection

The following data were retrieved from the patients’ medical records: parity, menopausal status at diagnosis, age at diagnosis, CA125 concentrations at the time of diagnosis and at the end of treatment, number of chemotherapy cycles and response, type of debulking surgery, debulking outcome, number of recurrences, type of treatment for the recurrences, vital status at the end of follow-up, DFS and OS. DFS was defined as time between final day of primary treatment until histological confirmation of recurrent disease or start of second-line therapy. Patients were censored if they had no recurrence by the 1 May 2020, or if they were deceased prior to the end of follow-up without evidence of a recurrence. For OS, death was recorded as an event while patients were censored at the end of follow-up (1 May 2020) or on the date of last contact if they chose to continue follow-up in a different hospital. Debulking outcome was classified as either ‘complete’ (i.e., no macroscopic disease), ‘optimal’ (i.e., macroscopic residue < 1 cm) or ‘incomplete’ (i.e., macroscopic disease > 1 cm) [18]. Data on menopausal status was lacking for ten patients below the age of 55. For these patients, endometrial sections were reviewed by an expert gynecological pathologist (S.L.B.) and menopausal status was determined based on atrophy of the endometrium.

### 2.3. Sample Preparation, mRNA Extraction and RT-qPCR Analysis

Original histological sections of HGSC samples were reviewed by one of two expert gynecological pathologists (S.L.B. and M.H.F.M.L.-B.). Representative sections with sufficient tumor cells were annotated and HGSC samples containing < 40% tumor cells were excluded from analysis to minimize stromal contamination. Five-micrometer formalin-fixed paraffin-embedded (FFPE) sections of primary HGSC samples were cut with a microtome (RM2255, Leica Biosystems, Wetzlar, Germany). Depending on total annotated tumor area, multiple sections were scraped manually resulting in at least 20 mm^2^ tumor surfaces. Total mRNA was isolated following the manufacturer’s protocol (VERSANT^®^ Tissue Preparation Reagents kit, Siemens, Erlangen, Germany) and mRNA concentrations were measured using the Qubit^®^ RNA HS Assay Kit and Qubit^®^ Fluorometer (Invitrogen, Thermo Fisher Scientific, Waltham, MA, USA). Pathway-specific target gene expression levels were measured by RT-qPCR using the SuperScript^TM^ III Platinum^TM^ One-Step qRT-PCR kit (Invitrogen, Thermo Fisher Scientific, USA), PCR plates (OncoSignal, Philips MPDx, Eindhoven, The Netherlands) and a CFX96 Real-Time PCR Detection System (BioRad, Hercules, CA, USA).

### 2.4. OncoSignal Pathway Assay

Anonymized RT-qPCR data were provided to Philips Research to determine functional STP activity using previously described knowledge-based probabilistic Bayesian computational pathway models [10,12,13]. An important and unique advantage of the commercially available pathway activity assays is that, in principle, they can be performed on every cell or tissue type. Alternative approaches often require fresh or fresh frozen tissue samples and may be limited by the requirement of a large amount of tissue [19]. OncoSignal pathway models are developed and validated using Affymetrix microarray expression data [10]. The models infer activity of the corresponding transcription factor complex from the expressions of pathway-specific target genes. To facilitate the use of RT-qPRC data obtained from FFPE samples, the models were adapted based on a selection of the most informative pathway-specific target genes. STP activity of the following pathways was determined: androgen receptor (AR), estrogen receptor (ER), phosphoinositide-3-kinase (PI3K), Hedgehog (HH), transforming growth factor beta (TGF-β) and the canonical wingless-type MMTV integration site (Wnt) pathway. The selected target genes included in the original models have been described in detail previously: the ER and Wnt pathways [10]; the AR, HH and TGF-β pathways [13] and the PI3K–FOXO pathway [13,20]. Activity scores represent the likelihood of a certain pathway being active, where 0 corresponds to the probability of an inactive pathway and 100 to the probability of an active pathway, as described previously [14]. For PI3K, the pathway activity is determined based on Forkhead Box Protein O (FOXO) transcription factor activity as they are directly inversely related in the absence of cellular oxidative stress [13]. To assess cellular oxidative stress, mRNA expression levels of superoxide dismutase 2 (*SOD2*), a FOXO target gene, were used. There are a few important considerations for the interpretation of the generated pathway activity scores: 1. the pathway activity score range (minimum-maximum activity) on the normalized scale is unique for each cell or tissue type. Once the range has been defined using samples with known pathway activity, the absolute value for every new sample can be directly interpreted against that reference. If the range has not been defined, differences in pathway activity scores between samples can be interpreted; 2. in the same sample the pathway activity scores of different signaling pathways cannot be compared, since each of the signaling pathways has its own range of activity scores; and 3. pathway activity scores are highly quantitative, and even small differences can be reproducible and meaningful. In addition to pathway-specific target genes, we determined the expression levels of SOD2 and KI-67 as a marker for cellular oxidative stress and cell proliferation, respectively.

### 2.5. Statistical Analysis

For clinicopathological characteristics, normally distributed continuous variables are presented as mean values with standard deviation (SD) and compared with a *t*-test. Skewed continuous variables are presented as the median with interquartile range (IQR) and compared with a Mann–Whitney *U* test or Kruskal–Wallis test. Categorical variables are presented as frequencies with percentage and compared with a Fisher’s exact test. STP activity per survival group is presented as boxplots displaying the median and IQR with overlying dot plots representing individual patient samples. Univariate Cox proportional hazards regression analysis was used to assess possible correlations between pathway activity and DFS and OS. Given the differences in hormonal status, we performed separate analyses for premenopausal and postmenopausal women. Subgroups were formed based on ER pathway activity, dividing the sample set in quartiles. Boxplots displaying pathway activity per ER subgroup were generated. DFS and OS of the newly formed subgroups were visualized in Kaplan–Meier curves and log-rank tests were used to test for differences. Subsequently, samples were divided over three subgroups containing: 1. Samples with ER pathway activity in the lowest quartile and PI3K pathway activity in the highest quartile; 2. Samples with ER pathway activity in the highest quartile and PI3K pathway activity in the lowest quartile; and 3. The remaining samples. Boxplots were created to visualize pathway activity per subgroup. *p*-values < 0.050 were considered statistically significant. Basic statistical analyses were performed using SPSS (IBM SPSS Statistics, version 26) and data visualization was conducted using Rstudio (Rstudio, Inc. version 1.1.463).

### 2.6. Ethical Approval

Due to the retrospective nature of the study, the Medical Research Involving Human Subjects Act (Dutch: Wet Medisch-wetenschappelijk Onderzoek met Mensen) does not apply, which was confirmed by the Medical research Ethics Committees United (MEC-U, study number W16.108). Given that a majority of patients had passed away by the time of inclusion and that our analysis would not yield any outcome of interest to either the patient or their families, patient approval was waived by local hospital committees under the condition, that prior to inclusion, medical files were checked for any signs that a patient would disapprove of the use of residual bodily tissue.

## 3. Results

### 3.1. Study Population

We identified 580 patients with advanced stage HGSC in the NCR database treated in our region, 157 of which met our eligibility criteria. Thirty-five patients with a DFS between 12 and 24 months were excluded. For the remaining 122 patients, histological sections of the primary tumor were retrieved. Review of the histological sections resulted in the exclusion of 36 women as the samples contained less than 40% tumor cells. Thus, 86 primary tumor samples were available for analysis of STP activity. Internal quality control resulted in the exclusion of one more patient due to insufficient mRNA concentration. STP activity results of 85 patients were included in our analysis, of which 52 were in the short DFS group and 33 in the long DFS group. There were no differences between the two groups in age at diagnosis, parity, menopausal status at diagnosis or FIGO stage. In the long DFS group, we found lower CA125 concentrations at diagnosis (*p* = 0.003) and after treatment (*p* = 0.027), as well as a higher number of primary debulking surgeries (*p* = 0.007) and complete debulking outcomes (*p* = 0.033). Furthermore, the number of recurrences was lower in the long DFS group (*p* < 0.001). An overview of clinicopathological characteristics per group is presented in Table 1.

### 3.2. Signal Transduction Pathway Activity in the Short and Long DFS Groups

For 85 samples, we determined activity of the AR, ER, PI3K, HH, TGF-β and Wnt pathways. For two samples there were high SOD2 levels which provided evidence of cellular oxidative stress, indicating that the PI3K pathway activity may be underrepresented and thus should be interpreted with caution. These samples are clearly marked in the figures. When comparing the short- and long DFS groups, no significant differences between median STP activity of the abovementioned pathways were found (Figure 1). In both survival groups, we observed a wide variety in STP activity among individual samples, mainly for the PI3K, TGF-β and Wnt pathways.

Our cohort included both pre- and postmenopausal women. As menopausal status may affect the availability of androgens and estrogens and therefore AR and ER pathway activity, univariate Cox proportional hazards regression analysis were used to assess the effect of menopausal status on the relation between pathway activity and survival. In premenopausal women (*n* = 16), none of the pathways were significantly related to OS or DFS. In postmenopausal women (*n* = 67), ER pathway activity was associated with favorable DFS (Hazard Ratio (HR) = 0.943; 95% confidence interval (95%CI) = 0.894 to 0.995; *p* = 0.033) and OS (HR = 0.930; 95%CI 0.868 to 0.997; *p* = 0.041). Results are visualized in Figure 2.

To investigate the relation between survival and ER pathway activity in postmenopausal women, this subgroup was divided into quartiles based on ER pathway activity. Samples in quartile 1 had an ER pathway activity score ranging from 0.12 to 4.80 (median 1.33), for quartile 2 ER scores ranged from 5.88 to 9.87 (median 9.23), for quartile 3 from 9.90 to 12.14 (median 11.46) and for quartile 4 from 12.23 to 27.94 (median 14.80). Survival analysis revealed a difference in both DFS and OS among the quartiles (log-rank *p* = 0.006 and *p* < 0.001, respectively) with the shortest DFS and OS for patients with ER pathway activity in the lowest quartile. Kaplan–Meier curves of DFS and OS per quartile are shown in Figure 3. There were no significant differences in baseline characteristics among the quartile groups that may influence the difference in survival such as CA125 concentration at diagnosis, debulking outcome or CA125 concentration after complete treatment. An overview of clinicopathological characteristics of the ER subgroups is provided in Appendix A. We hypothesized that the association between ER pathway activity and survival might have been influenced by the activity of other pathways. Figure 4 provides an overview of pathway activity per ER subgroup. Comparing median STP activity of the remaining pathways did not reveal significant differences. However, the subgroup with the lowest ER pathway activity scores was characterized by higher PI3K pathway activity when compared to the other subgroups. Inversely, the subgroup containing samples with the highest ER pathway activity scores was characterized by lower PI3K pathway activity.

To further investigate the inverse relation between ER and PI3K pathway activity, we divided samples of postmenopausal women in groups based on ER and PI3K pathway activity. Subgroup 1 contained all samples with ER pathway activity in the lowest quartile and PI3K pathway activity in the highest quartile (*n* = 6), subgroup 2 contained all samples with ER pathway activity in the highest quartile and PI3K pathway activity in the lowest quartile (*n* = 6) and subgroup 3 contains all remaining samples (*n* = 55). For subgroup 1, ER pathway activity scores ranged from 0.36 to 4.80 (median 0.91) and PI3K pathway activity scores ranged from 62.83 to 89.17 (median 75.39). For subgroup 2, ER scores ranged from 12.86 to 27.94 (median 18.22) and PI3K scores from 33.28 to 36.82 (median 36.01) and for subgroup 3, ER scores ranged from 0.12 to 25.40 (median 9.87) and PI3K scores ranged from 19.44 to 84.23 (median 44.59). Figure 5 provides an overview of pathway activity per subgroup. When comparing pathway activity among the subgroups, there was a difference in AR (*p* = 0.009) and TGF-β pathway activity (*p* = 0.041). Subgroup 1 was associated with low AR and TGF-β pathway activity compared to the other subgroups, while subgroup 2 was characterized by higher TGF-β pathway activity. There was no statistically significant difference in expression levels of the KI-67 proliferation marker among the subgroups, neither were there differences in DFS or OS.

Subsequently we repeated the analysis including both pre- and postmenopausal women. Again, samples were divided based on low ER and high PI3K pathway activity (subgroup A, *n* = 9), high ER and low PI3K pathway activity (subgroup B, *n* = 6) and the remaining samples (subgroup C, *n* = 70). For subgroup A, ER pathway activity scores ranged from 0.22 to 4.80 (median 1.05) and PI3K pathway activity scores ranged from 60.61 to 89.17 (median 73.74). For subgroup B, ER scores ranged from 14.82 to 27.94 (median 21.16) and PI3K scores from 31.31 to 36.82 (median 34.51) and for subgroup C, ER scores ranged from 0.12 to 48.64 (median 10.72) and PI3K scores ranged from 19.44 to 84.23 (median 44.59). Figure 6 shows an overview of pathway activity per subgroup. The difference in AR (*p* = 0.001) and TGF-β (*p* = 0.001) was retained; furthermore, there was a difference in Wnt (*p* = 0.018) pathway activity between the subgroups.

To summarize, we identified two subgroups based on ER and PI3K pathway activity which were characterized by high versus low AR and TGF-β pathway activity in postmenopausal women. When premenopausal women were included, there also was a difference in Wnt pathway activity among the subgroups.

## 4. Discussion

In this exploratory study, we assessed whether STP activity can explain differences in survival in HGSC patients. We analyzed STP activity in 85 primary tumor samples of patients diagnosed with advanced stage HGSC who achieved complete remission after treatment and a DFS below 12 months (short DFS) or over 24 months (long DFS). There were no differences between these two groups in median AR, ER, PI3K, HH, TGF-β and Wnt pathway activity. Since we observed a wide variety in activity of several STPs in both short and long DFS groups, our division of HGSC in two groups may have precluded the discovery of more subtle interactions between pathway activity and survival. The wide variety of STP activity may also indicate the existence of more specific subgroups. In univariate Cox proportional hazards analysis, stratification for menopausal status revealed a positive correlation between ER pathway activity and both DFS and OS in postmenopausal women. Moreover, Kaplan–Meier survival analysis demonstrated a difference in both DFS and OS among subgroups based on ER pathway activity quartiles in postmenopausal women. The difference is mostly due to the lowest quartile compared to the others as it was characterized by the shortest DFS and OS. Within each of the other quartiles, there were large differences in DFS and OS, for example in quartile 2 which had the second shortest DFS but the longest OS. The differences between DFS and OS within the clusters are illustrative of differences in intrinsic behavior. Alternatively, the low ER pathway activity in quartile 1, while there was no evident difference in survival among the second, third and fourth quartiles, may also suggest that an inactive pathway in particular is negatively related to survival. The comparability in survival among the higher quartiles may result from a lack of samples with a particularly active ER pathway or may indicate that whether or not the ER pathway is active is more important than the actual level of activity. While these samples may be active compared to other HGSC samples, ER pathway activity is still low to moderate when compared to healthy Fallopian tube tissue [21]. Thus, normal ER pathway activity, which is necessary for differentiated cell functions in healthy cells, is lost in HGSC. Alternatively, the differences in survival may result from preferential activity of the ER-α transcription factor over the ER-β transcription factor [22,23]. ER-β mediated signaling is tumor suppressing while ER-α mediated signaling results in increased proliferation and thus acts as a tumor promotor. Preferential signaling may thereby contribute to a tumor-driving role of the ER pathway. A slight upregulation of ER-α in HGSC samples has been described previously [22]. It should be noted that the number of included women is small and diminishes over time; results should therefore be interpreted with caution.

Although individual studies on the prognostic role of ER protein expression have previously resulted in conflicting outcomes, a recent meta-analysis concluded that OS was unrelated to ER protein expression in serous ovarian carcinoma (HR 0.90; 95%CI 0.75 to 1.08) [24,25,26,27]. Unfortunately, due to a lack of suitable immunohistochemical antibodies, there is no reliable distinction between ER-α and ER-β expression. Furthermore, a direct comparison to our results is hindered as ER protein expression does not necessarily reflect an active ER signaling pathway [28]. None of the studies differentiate between pre- and postmenopausal women, while our findings suggest that hormonal status (e.g., pre- and postmenopausal) could influence tumor behavior. Climacteric changes in steroid hormone metabolism may alter the effect of hormone receptor pathway signaling on ovarian carcinogenesis. In premenopausal women, endocrine estrogen synthesis by the ovaries results in fluctuating levels of circulating estradiol (E2) [29]. In postmenopausal women, estrone (E1) is most abundant due to depletion of the ovarian function [30]. However, active E2 is synthesized from E1 in peripheral tissue such as adipocytes or by intracellular formation in estrogen-dependent tumor cells [31]. Our results suggest that, in a subgroup of postmenopausal women, the tumor is either insensitive to residual levels of estrogens or is unable to produce estrogens itself. As a consequence, inactivity of the ER signaling pathway may promote tumor progression to a more aggressive phenotype, resulting in poorer survival outcomes. Thus, this subset of postmenopausal women may benefit from high dosed estrogen replacement therapy under the condition that ER-β is the dominant receptor type. Alternatively, selective ER-β agonists might be required [22,23].

When forming new subgroups based on ER and PI3K pathway activity, we found that subgroup 1 was also characterized by the lowest AR pathway activity, which further supports the loss of normal pathway activity in HGSC which is required for normal differentiated cell functions. An interplay between these pathways has been reported in ovarian cancer previously. In the study of Martins et al., immunohistochemical ER, phosphatase and tensin homologue (PTEN), and AR protein expression was found in 3244 HGSC samples [32]. PTEN is a tumor suppressor of the PI3K pathway and loss of PTEN is associated with hyperactivation of the PI3K pathway activity. In line with our findings, positive PTEN protein expression was strongly correlated to both ER and AR protein expression [32]. Furthermore, a study in PTEN-deficient prostate cancer showed that AR and PI3K pathway activity were inversely related. Inhibition of AR signaling resulted in an upregulation of PI3K signaling and vice versa [33]. In our subgroups based on ER and PI3K pathway activity, we see an inverse relation between AR and PI3K pathway activity (Figure 5 and Figure 6). Hill et al. studied AR and PI3K in ovarian cancer and, although they reported some level of interaction, it was not reciprocal as is the case in prostate cancer [34]. This outcome seems typical for ovarian cancer research and may result from the inclusion of several histological subtypes of ovarian cancer or from intra-tumoral heterogeneity in which a single tumor constitutes of several cell populations with their own features and specific behaviors [35]. In their study, Hill et al. reported that the relation between the two pathways requires further studies as the outcome may also result from their choice of AR activity marker or the use of Metformin as PI3K inhibitor [34].

Another finding in our subgroups based on the inverse relation between ER and PI3K pathway activity is the low TGF-β pathway activity of subgroup 1. A crosstalk between the PI3K and TGF-β pathway has been described previously, in which the anti-proliferative effect of TGF-β signaling is decreased by PI3K pathway activation or even reversed to tumor promoting depending on the concomitant presence of an active MAPK-AP1 pathway [20]. Analysis of pre- and postmenopausal women also resulted in a difference in Wnt pathway activity among the subgroups. In subgroup A, we found low TGF-β and Wnt pathway activity, while in subgroup B both pathways appear to be active. An intricate cooperation of the TGF-β and Wnt pathways acts as a tumor-promotor, as described previously [36]. In contrast, the combination of Wnt pathway activity and FOXO transcription factor activity (i.e., an inactive PI3K pathway) acts as tumor suppressive in prostate cancer cells [37].

In a previously published study we used STP assays on a publicly available dataset of clinically annotated HGSC samples by Tothill et al. [38,39]. We applied a similar analysis to the Tothill dataset, with subgroups based on low ER and high PI3K pathway activity and high ER and low PI3K pathway activity compared to the remaining samples. Although the dataset contained ample clinical details, there were no data on menopausal status and thus we were unable to perform a subgroup analysis of postmenopausal women. Instead, we included all HGSC patients with a DFS below 12 or above 24 months. This analysis revealed comparable results for AR pathway activity (*p* = 0.002); the subgroup with low ER pathway activity and high PI3K pathway activity had low AR pathway activity and the subgroup with high ER pathway activity and low PI3K pathway activity had high AR pathway activity.

A major strength of our study is the clearly defined patient population. Ovarian carcinoma is a heterogeneous disease and a generally used term for several histotypes, each with their own distinct characteristics, course of disease and optimal treatment. To limit heterogeneity and treatment effects on our main outcome, DFS, we formulated concise in- and exclusion criteria. Although this approach has its limitations, as it results in the exclusion of a substantial number of patients, we feel it is justified and even necessary to answer our research question. As a result, the number of included patients is moderate to small, especially for the subgroup analysis. Future studies including larger groups of patients should be conducted to establish the translational value of our results. The effect of FIGO stage and treatment modalities are profound and thus may conceal the effects of differences in STP signaling. Another strength is the translation of real-time quantitative reverse transcription-PCR (RT-qPCR) results to pathway activity scores with potential clinical target and, in contrast to other tools to determine pathway activity, the STP assays used here can be used for individual patient samples and thus are suitable for use in daily clinical practice. This more personalized approach to determine tumor specific characteristics may also be beneficial for the selection of targeted therapies.

A limitation of our study is the retrospective nature, as we were dependent on the quality and tumor percentage of readily available samples and on medical files for patient characteristics. The hospitals from which patients were included were large referral centers for gynecologic oncology. This generally means that patients are in care of a nearby hospital and are only referred after a first round of diagnostics indicated an ovarian malignancy, lowering the availability of primary tumor samples. Patients may also choose to return to the referring hospital for adjuvant treatment and follow-up, limiting the availability of follow-up data for a small number of patients. In addition, we chose to exclude women with a DFS between 12 and 24 months to maximize possible differences between short- and long-term survivors. Although the number of excluded patients was limited (*n* = 35), and it is unknown for how many of these patients’ primary tumor samples were available for PCR analysis, this may have concealed subtler differences. Furthermore, the use of achieving complete remission as inclusion criterium has resulted in a selection bias in which patients with the poorest outcome were excluded.

## 5. Conclusions

While we have found a relation between survival and ER signaling pathway activity in advanced stage postmenopausal HGSC, much remains to be elucidated when it comes to STP activity in HGSC. Identification of patients with high risk of recurrence and poor survival could be particularly useful in stratification of patients to (maintenance) targeted therapies. In our assessment of STP activity of short- and long-term disease-free survivors of HGSC, we were unable to identify a single pathway responsible for the differences in survival. However, we were able to identify subgroups of patients which were characterized by high ER and AR pathway activity and low PI3K pathway activity and conversely low ER and AR pathway activity and high PI3K pathway activity.

## Figures and Tables

**Figure 1 cancers-13-05101-f001:**
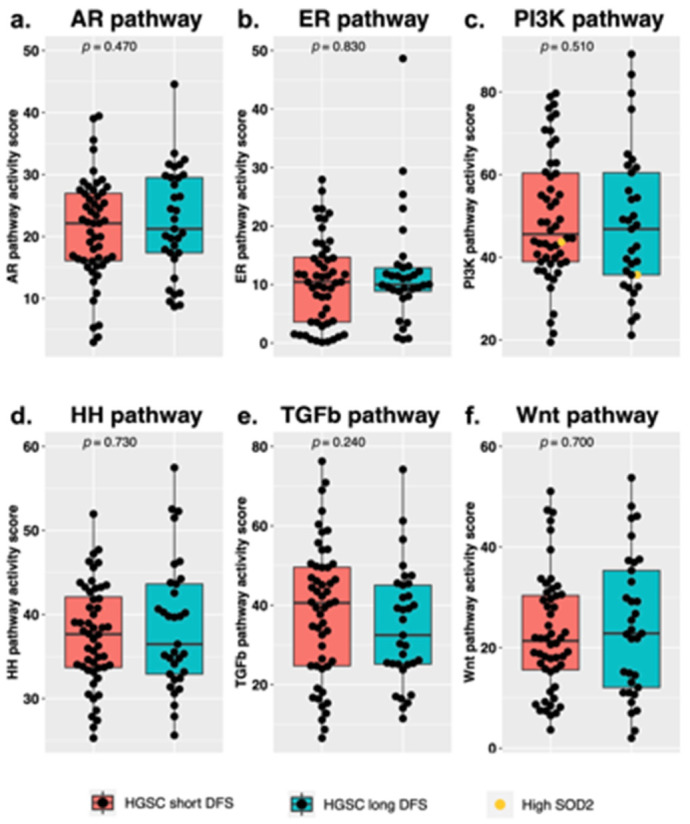
Signal transduction pathway activity measured in high-grade serous ovarian carcinoma (HGSC) samples in the short and long disease-free survival (DFS) groups, for the (**a**) androgen receptor (AR), (**b**) estrogen receptor (ER), (**c**) phosphoinositide-3-kinase (PI3K), (**d**) Hedgehog (HH), (**e**) transforming growth factor beta (TGF-β) and (**f**) canonical wingless-type MMTV integration site (Wnt) pathway. *p*-values were obtained from a Mann–Whitney *U* test. The samples with high SOD2 levels are marked in yellow. “HGSC short DFS” refers to a DFS below 12 months and “HGSC long DFS” refers to a DFS over 24 months.

**Figure 2 cancers-13-05101-f002:**
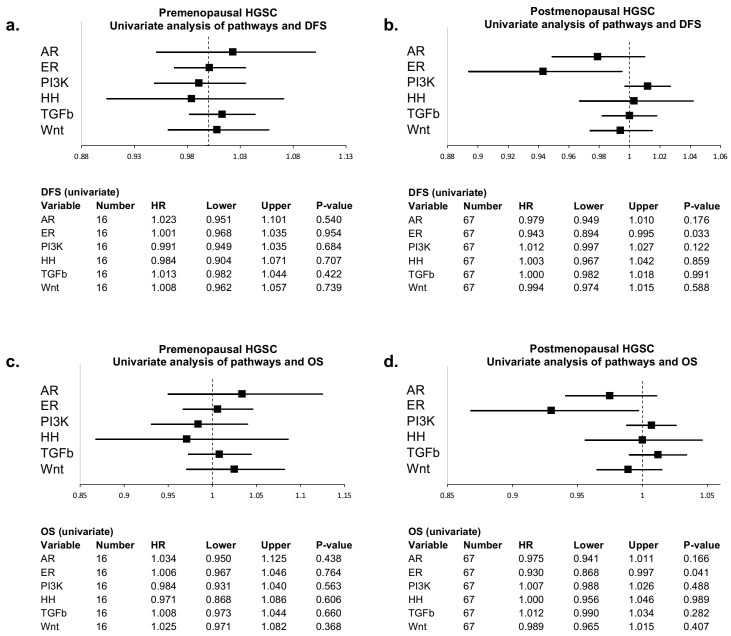
Forest plots of univariate Cox proportional hazards regression analysis of all pathways and survival. Hazard ratios (HR) with 95% confidence intervals are described for disease-free survival (DFS) in (**a**) premenopausal and (**b**) postmenopausal women, and overall survival (OS) in (**c**) premenopausal and (**d**) postmenopausal women.

**Figure 3 cancers-13-05101-f003:**
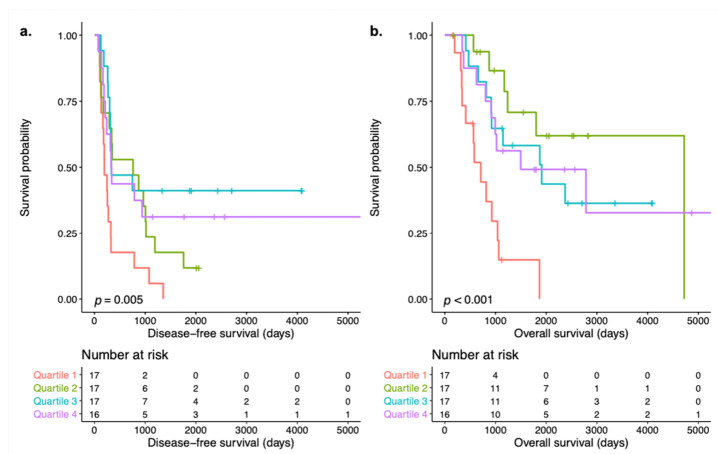
Kaplan–Meier survival analysis with log-rank tests and number at risk tables of subgroups based on quartiles of ER pathway activity in postmenopausal high-grade serous ovarian carcinoma. (**a**) Disease-free survival curves. (**b**) Overall survival curves.

**Figure 4 cancers-13-05101-f004:**
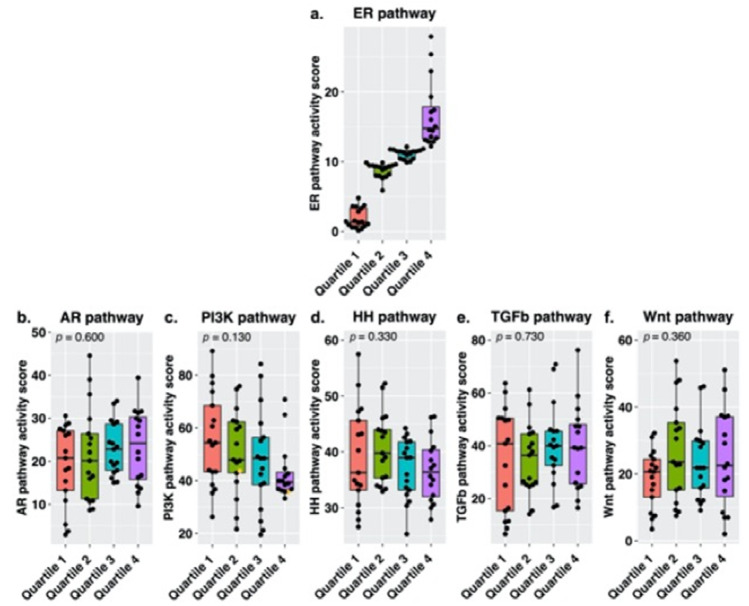
Signal transduction pathway activity measured in high-grade serous carcinoma samples of postmenopausal patients, for the (**a**) estrogen receptor (ER), (**b**) androgen receptor (AR), (**c**) phosphoinositide-3-kinase (PI3K), (**d**) Hedgehog (HH), (**e**) transforming growth factor beta (TGF-β) and (**f**) canonical wingless-type MMTV integration site (Wnt) pathway. Subgroups were created by dividing ER pathway activity into quartiles (Q1, Q2 and Q3 *n* = 17, Q4 *n* = 16). Median pathway activity was compared among the ER quartiles, *p*-values were derived using a Kruskal–Wallis test. The samples with high SOD2 levels are marked in yellow.

**Figure 5 cancers-13-05101-f005:**
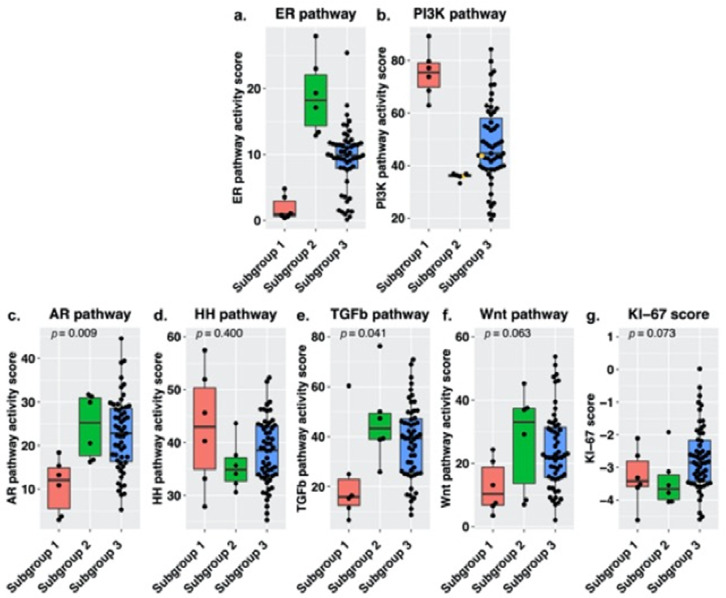
Signal transduction pathway activity measured in the different subgroups containing high-grade serous carcinoma samples of postmenopausal women, for the (**a**) estrogen receptor (ER), (**b**) phosphoinositide-3-kinase (PI3K), (**c**) androgen receptor (AR), (**d**) Hedgehog (HH), (**e**) transforming growth factor beta (TGF-β) (**f**) canonical wingless-type MMTV integration site (Wnt) pathway and (**g**) KI-67 expression levels. Subgroups were based on ER and PI3K pathway activity. Subgroup 1 (*n* = 6) contains samples with low ER pathway activity (quartile 1) and high PI3K pathway activity (quartile 4). Subgroup 2 (*n* = 6) contains samples with high ER pathway activity (quartile 4) and low PI3K pathway activity (quartile 1). Subgroup 3 (*n* = 55) contains the remaining samples. Median pathway activity was compared among the subgroups, *p*-values were derived using a Kruskal–Wallis test. The samples with high SOD2 levels are marked in yellow.

**Figure 6 cancers-13-05101-f006:**
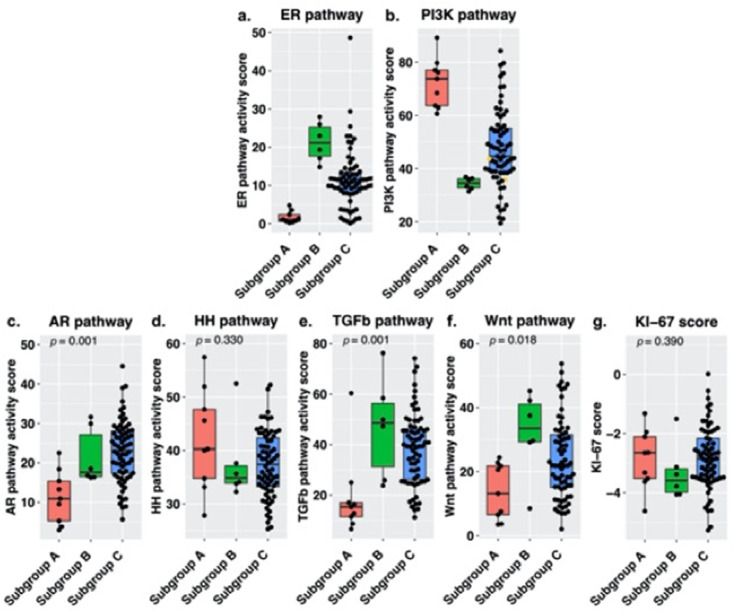
Signal transduction pathway activity measured in the newly formed subgroups containing high-grade serous carcinoma samples of both pre- and postmenopausal women, for the (**a**) estrogen receptor (ER), (**b**) phosphoinositide-3-kinase (PI3K), (**c**) androgen receptor (AR), (**d**) Hedgehog (HH), (**e**) transforming growth factor beta (TGF-β) (**f**) canonical wingless-type MMTV integration site (Wnt) pathway and (**g**) KI-67 expression levels. Subgroups were based on ER and PI3K pathway activity. Subgroup A (*n* = 9) contains samples with low ER pathway activity (quartile 1) and high PI3K pathway activity (quartile 4). Subgroup B (*n* = 6) contains samples with high ER pathway activity (quartile 4) and low PI3K pathway activity (quartile 1). Subgroup C (*n* = 70) contains the remaining samples. Median pathway activity was compared among the subgroups, *p*-values were derived using a Kruskal–Wallis test. The samples with high SOD2 levels are marked in yellow.

**Table 1 cancers-13-05101-t001:** Clinicopathological characteristics of the included women diagnosed with advanced stage high-grade serous ovarian carcinoma. Women were divided into two groups based on short (<12 months) and long (>24 months) disease-free survival (DFS).

Variation	Short DFS *n* = 52 (%)	Long DFS *n* = 33 (%)	*p*-Value *
**Age at diagnosis**			0.856
Mean (SD)	62 (12)	61 (12)	
**Parity**			0.477
0	7 (13)	7 (21)	
1–2	20 (38)	16 (48)	
≥3	17 (33)	8 (24)	
Missing	8 (15)	2 (6)	
**Menopausal status**			1.000
Premenopausal	10 (19)	6 (18)	
Postmenopausal	40 (77)	27 (82)	
Missing	2 (4)	0 (0)	
**FIGO stage**			0.758
IIIC	43 (83)	29 (88)	
IV	9 (17)	4 (12)	
**CA125 concentration at diagnosis**			0.003
Median (IQR)	657 (258–2125)	244 (120–415)	
Missing	2	2	
**CA125 concentration after treatment**			0.027
Median (IQR)	13 (9–17)	10 (6–14)	
Missing	12	3	
**Debulking type**			0.007
Primary	21 (40)	24 (73)	
Interval	26 (50)	9 (27)	
Other **	5 (10)	0 (0)	
**Debulking outcome**			0.033
Complete (no macroscopic residue)	30 (58)	27 (82)	
Optimal (residue <1cm)	9 (17)	5 (15)	
Incomplete (residue >1cm)	12 (23)	1 (3)	
Missing	1 (2)	0 (0)	
**Number of recurrences**			<0.001
No recurrence	0 (0)	16 (49)	
1	40 (76)	14 (42)	
2	6 (12)	1 (3)	
≥3	6 (12)	2 (6)	
**Disease-free survival (days)**			<0.001
Median (IQR)	195 (128–297)	1192 (952–2210)	
**Overall survival (days)**			<0.001
Median (IQR)	704 (425–991)	2058 (1618–2804)	

* Differences in continuous variables were tested with a *t*-test (normal distribution) or Mann–Whitney *U* test (skewed distribution). For categorical variables, *p*-values were obtained from a Fisher’s exact test. ** Incomplete primary debulking followed by neo-adjuvant chemotherapy and afterwards a secondary interval debulking with adjuvant chemotherapy. Abbreviations: SD, standard deviation; IQR, interquartile range.

## Data Availability

The datasets generated during and/or analyzed during the current study will be available on reasonable request.

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
