# Peer review of "Survival Is Related to Estrogen Signal Transduction Pathway Activity in Postmenopausal Women Diagnosed with High-Grade Serous Ovarian Carcinoma"

_cancers, 2021, doi:10.3390/cancers13205101_

Round 1
Reviewer 1 Report
This paper is well written and clearly describes the results of STP analysis in a well defined group of late stage ovarian cancer patients. STP scores are correlated to clinical outcome in terms of disease free survival.
GENERAL COMMENTS
- Usually one relates the consequential result to a (causal) factor. So, as a matter of principle, it would be more logic to speak of ‘survival related to pathway activity’ rather than the other way around. Use of the more neutral ‘correlate’ could be another option.
- Although the paper clearly advertises the advantages and opportunities offered by the pathway activity assay used (OncoSignal), it is less clear from the description how the pathway activity scores are actually obtained. A brief expansion on the computational pathway models described in the references provided would be helpful.
- It should (also) be discussed and kept in mind that the authors for all good intentions and reasons limited this study to patients with initial complete remission, as this constitutes an already favourable subpopulation.
MAJOR COMMENTS
L.110 The arguments for selection criteria to determine ‘long DFS’ vs ‘short DFS’ are not very strong. Firstly, the DFS s on which they base the choice of cut-off are derived from one single Italian study. Other and larger studies could have been cited with median DFS upto 30 months (Kurta e.a., JCO 2014). Secondly, the ’16 to 19 months’ is not a range, but actually two median DFSs for two distinct groups in that Italian study (primary vs interval debulking, Ghirardi e.a., Gynecol Oncol 2020). Thirdly, the range of DFS to exclude is apparently not chosen on basis of percentiles as the lower cut-off was chosen 3 months below the lower DFS from the Italian study, and the upper cut-off 5 months above the higher DFS in that study (unless distribution was uneven).
L.147 Many other assays, besides pathway activity assays can be performed on all cell or tissue type. Please explain, detail or rephrase why this would be unique for pathway activity assays
L.164 As any new sample can (or should ?!) be interpreted against a reference, in what way have which references been obtained and used for this study ?
L.223 The outcome of surgery is reported in a slightly confusing fashion as since 2010 ‘optimal debulking’ has been defined as no visible tumour (Stuart e.a., Int J Gynecol Cancer 2011). Since there is also a ‘complete’ debulking group, it is unclear from the table how these outcomes are defined. If for the purpose of this study ‘optimal’ has been defined on basis of a cut-off of residual tumour size this should be revised to comply with current outcome definitions.
L.337 The subgroup survival analysis is extensively discussed, however the incoherent survival differences between the subgroups need to be cautiously interpreted and are most likely due to the extremely low numbers at risk in the Kaplan-Meier curves. These data by themselves certainly do not justify the hypotheses made in this paragraph.
MINOR COMMENTS
L.37 In the summary a range of percentages of DFS is mentioned that is actually not a range (see earlier comment)
L.64 The sentence on individual applicability of clinical prognostic factors reads difficult but also does not correctly reflect the conclusion of the referred paper. Reword more cautiously.
L.127 Rather than advertising the pathologist as an ‘expert’, histologic criteria should be given on basis of which menopausal status was determined.
L.380 Why do the authors report that the ER/P13K subgroup characteristics correspond to the definition on which they based these subgroups ?! Maybe the authors ‘defined’, rather than ‘found’ these characteristics?
Reviewer 2 Report
The research manuscript submitted by Van Lieshout et al. identifies a prognostic signature for the clinical outcome of ovarian cancer patients.
Overall, the methodologies used are appropriated and inclusion/exclusion criteria are well described, the results discussed are clear and the points of streght/weakness well addressed.
The only concern is about the low number of the patients' cohort, that may be extended in future studies to better dissect the translational value of the signature proposed.
Reviewer 3 Report
The manuscript “Estrogen signal transduction pathway activity is related to survival in postmenopausal women diagnosed with High-Grade Serous Ovarian Carcinoma” by Laura van Lieshout and co-authors investigate possible relations between signal transduction pathway (STP) activity and survival. Authors assessed the functional activity of the androgen receptor (AR), estrogen receptor (ER), phosphoinositide-3-kinase (PI3K), Hedgehog (HH), transforming growth factor beta (TGF-?) and canonical wingless-type MMTV integration site (Wnt) pathway in 85 primary tumor samples of patients with FIGO stage IIIC to IVB HGSC and a DFS below 12 (n=52) or over 24 months (n=33). There were no significant differences in median pathway activity between patients with a short and long DFS. In univariate Cox proportional hazards analysis, ER pathway activity was related to a favorable DFS and OS in postmenopausal women (p=0.033 and p=0.041, respectively), but not in premenopausal women. Authors divided the postmenopausal group in subgroups based on ER pathway activity quartiles. Survival analysis revealed that postmenopausal women in the lowest ER quartile had a shorter DFS and OS (log-rank p=0.006 and p<0.001, respectively). Furthermore, authors were able to form subgroups of patients based on an inverse relation between ER and PI3K pathway activity. In conclusion, in postmenopausal patients with advanced stage HGSC, low functional ER pathway activity was associated with a poorer survival outcome. However, there are concerns which must be taken into account before the work can be reconsidered for publication.
- Can author explain why exclude 35 patients with DFS between 12 and 24 months?
- Can you provide the detailed information about RT-qPCR of target genes, eg primer?
- Figure 3: How to divide quartile? Please provide the cut-off value. Figure 3A: Please check P value.
- Figure 4: How to divide quartile? Please provide the cut-off values.
- Figures 5 and 6: Please provide the cut-off values of ER and PI3K.
Round 2
Reviewer 3 Report
The revised manuscript “Survival is related to estrogen signal transduction pathway activity in postmenopausal women diagnosed with High-Grade Serous Ovarian Carcinoma” have adequately addressed my previous concerns and the paper is now acceptable for publication.